# Identification of Novel Metabolic Subtypes Using Multi-Trait Limited Mixed Regression in the Chinese Population

**DOI:** 10.3390/biomedicines10123093

**Published:** 2022-12-01

**Authors:** Kexin Ding, Zechen Zhou, Yujia Ma, Xiaoyi Li, Han Xiao, Yiqun Wu, Tao Wu, Dafang Chen

**Affiliations:** Department of Epidemiology and Biostatistics, School of Public Health, Peking University, Beijing 100191, China

**Keywords:** metabolic subtype, metabolic risk factor, cardiovascular disease, genetic basis, Chinese

## Abstract

The aggregation and interaction of metabolic risk factors leads to highly heterogeneous pathogeneses, manifestations, and outcomes, hindering risk stratification and targeted management. To deconstruct the heterogeneity, we used baseline data from phase II of the Fangshan Family-Based Ischemic Stroke Study (FISSIC), and a total of 4632 participants were included. A total of 732 individuals who did not have any component of metabolic syndrome (MetS) were set as a reference group, while 3900 individuals with metabolic abnormalities were clustered into subtypes using multi-trait limited mixed regression (MFMR). Four metabolic subtypes were identified with the dominant characteristics of abdominal obesity, hypertension, hyperglycemia, and dyslipidemia. Multivariate logistic regression showed that the hyperglycemia-dominant subtype had the highest coronary heart disease (CHD) risk (OR: 6.440, 95% CI: 3.177–13.977) and that the dyslipidemia-dominant subtype had the highest stroke risk (OR: 2.450, 95% CI: 1.250–5.265). Exome-wide association studies (EWASs) identified eight SNPs related to the dyslipidemia-dominant subtype with genome-wide significance, which were located in the genes *APOA5*, *BUD13*, *ZNF259*, and *WNT4*. Functional analysis revealed an enrichment of top genes in metabolism-related biological pathways and expression in the heart, brain, arteries, and kidneys. Our findings provide directions for future attempts at risk stratification and evidence-based management in populations with metabolic abnormalities from a systematic perspective.

## 1. Introduction

The aggregation and interaction of metabolic risk factors within an individual constitutes a complex and progressive pathophysiological state, which substantially increases the risk of developing type 2 diabetes (T2D) and cardiovascular disease (CVD) [1]. These risk factors, including obesity, hypertension, hyperglycemia, dyslipidemia, pro-inflammatory states, and pro-thrombotic states, affect individuals in various numbers, types, and degrees, resulting in highly heterogeneous pathogeneses, manifestations, and outcomes. Unfortunately, major challenges remain in the risk stratification and targeted management of large populations with metabolic abnormalities precisely due to the unrevealed heterogeneity [2].

In an attempt to deconstruct the heterogeneity, recent studies have performed cluster analyses of phenotypic traits to identify complex disease subtypes [3,4,5]. Dahl et al. proposed reverse GWASs (RGWASs) to infer subtypes by clustering multiple traits with a finite-mixture-of-regressions method called multi-trait limited mixed regression (MFMR), which was designed specifically for large, multi-trait datasets [6]. By carefully incorporating covariates to remove heterogeneity due to confounders, such as age and population structure, RGWASs increase the odds that discovered subtypes are biologically meaningful [7]. RGWASs were then applied to identify three novel metabolic subtypes with different T2D and coronary heart disease (CHD) risks, yet the diseases themselves were involved as binary cluster traits. In addition, there have been growing efforts to explore and characterize the metabolic landscape of tumors using unsupervised cluster analysis of metabolic traits [8,9,10,11,12,13]. However, the direction of these studies tends to be disease-outcome-driven rather than metabolic-risk-factor-driven, treatment-oriented rather than prevention-oriented, which runs counter to advances in precision medicine and fails to maximize cost-effectiveness.

The increasing scourge of cardiometabolic diseases emphasizes the need to recognize those at risk in a timely manner and identify subtypes on the basis of underlying pathophysiologies and predispositions to adverse consequences in order to more effectively direct preventive and therapeutic strategies. Indeed, five possible pathologically relevant subtypes were proposed in theory by the CardioMetabolic Health Alliance, including an adiposity-dominant type, an insulin-resistance-dominant type, a vascular-dominant type, a lipid-dominant type, and an other-risk-factors-dominant type [2]. Further validations and optimized classifications of metabolic subtypes are urgently required in practice. In addition, genomic association studies offer insights into the underlying genetic heterogeneity suggested by the phenotypic variation observed.

To sum up, based on a community observational study in Beijing, in this study we sought to (1) infer metabolic subtypes, clustering individuals into subtypes according to their levels of multidimensional metabolic indicators using MFMR; (2) examine metabolic subtypes, testing whether the inferred subtypes have differential risks of CHD and stroke; and (3) explore the underlying genetic bases of the inferred subtypes.

## 2. Materials and Methods

### 2.1. Study Participants

This study was based on the Fangshan Family-Based Ischemic Stroke Study (FISSIC) [14]. FISSIC is an ongoing community observational study that started in June 2005 in the Fangshan District, a rural area southwest of Beijing, China. We used the baseline data from phase II of the FISSIC from December 2011 to April 2012, in which 9540 participants aged ≥40 years were recruited either in response to recruitment posters or by phone calls if their health records were available at a community health center [15].

This study was approved by the Ethics Committee of the Peking University Health Science Center (approval number: IRB00001052–13027), and all participants signed an informed consent form.

### 2.2. Data Collection

In the FISSIC study, questionnaires were administered to assess demographics, socioeconomic status, lifestyle factors, and medical and medication history. Smoking status was classified as current smoker or non-smoker (including never smoked and former smoker), and drinking status was also recorded. Medical comorbidities included hypertension, T2D, dyslipidemia, MetS and its five components, CHD, and stroke. Medication history of antihypertensive drugs, antidiabetic drugs, or lipid-lowering drugs in the past 3 days was also collected. Anthropometric measures, including height, weight, blood pressure (BP), and waist circumference, were collected for each participant. Body mass index (BMI) was calculated as kg/m^2^. Systolic blood pressure (SBP) and diastolic blood pressure (DBP) were measured in sitting position after resting for at least 5 min, and the averages of three consecutive measurements were used for analysis. Venous blood samples were taken from participants after 12 h of fasting. Serum or plasma samples were separated by centrifugation, typically within 30 min, and stored at −80° centigrade, then used for measurement of fasting blood glucose (FBG); 2 h postprandial blood glucose (2h-PBG) on a standard 75 g oral glucose tolerance test; serum lipid levels, including triglycerides (TGs), total cholesterol (TC), high-density lipoprotein cholesterol (HDL-C), and low-density lipoprotein cholesterol (LDL-C); and DNA analysis. All procedures were carried out by qualified professionals in accordance with regulations.

### 2.3. Disease Definition

Hypertension was defined as self-reported hypertension, antihypertensive treatment, or a blood pressure of at least 140 mmHg for systolic or 90 mmHg for diastolic blood pressure. T2D was defined as self-reported type 2 diabetes, FBG ≥ 7.0 mmol/L, 2h-PBG ≥ 11.1 mmol/L, or use of antidiabetic medication. Dyslipidemia was defined as self-reported history of hyperlipidemia or TC ≥ 5.18 mmol/L or TGs ≥ 1.7 mmol/L or LDL-C ≥ 3.37 mmol/L. CHD and stroke were defined as a self-reported history of physician-diagnosed CHD or stroke. The criterion for the diagnosis of metabolic syndrome was according to the National Cholesterol Education Program’s Adult Treatment Panel III (NCEP-ATP III) [16]. The criteria for clinical diagnosis of MetS are 3 or more of the following: (1) waist circumference ≥ 90 cm in men and 85 cm in women; (2) TGs ≥ 150 mg/dl; (3) HDL-C < 40 mg/dl; (4) blood pressure ≥ 130/85 mmHg; and (5) fasting glucose ≥ 110 mg/dl.

### 2.4. Genotyping and Quality Controls

DNA was extracted using a LabTurbo 496-Standard System (TAIGEN Bioscience Corporation, Taiwan, China), and the purity and concentration of DNA were measured using ultraviolet spectrophotometry. In addition, the genomic DNA sample was genotyped using a customized whole-exome chip based on the Human Exome BeadChip (Illumina, Inc., San Diego, CA, USA) (http://genome.sph.umich.edu/wiki/Exome_Chip_Design, accessed on 30 September 2022). The Infinium Human Exome BeadChip contained approximately 250,000 single nucleotide polymorphisms (SNPs) in protein-coding regions, but most of them came from European populations and could not cover the specific mutations of Asian populations well. Therefore, an additional 60,000 SNPs were designed on this basis. The increased SNPs were peculiar to Asian populations or previously reported in genome-wide association studies (GWASs) for such factors as blood pressure, lipids, and CHD [17].

Two negatives (blanks) and three positive controls were used to control the quality of the genotyping process, and the results were satisfactory. We also randomly selected 5% of the samples for repeat analysis to validate genotyping procedures. Plate-, individual-, and variant-level checks were conducted to exclude poor-quality genotype calls from the dataset. The individual-based quality control criteria included a call rate of < 99%, gender mismatch, excess heterozygosity, and relatedness. Variant-level quality control was performed to exclude variants with low cluster scores, low call rates (<99.9%), and those that deviated from Hardy–Weinberg equilibrium (*p* < 1 × 10^−4^). Qualified samples and genotypes were phased and imputed with SHAPEIT (segmented haplotype estimation and imputation tool) v2 [18,19] and IMPUTE v2 [20], respectively, using default parameters and the 1000 Genomes Project Phase III database (released in October, 2014) as the reference.

### 2.5. Statistical Analysis

#### 2.5.1. Clustering of Subtypes Using Multi-Trait Limited Mixed Regression

To deconstruct the heterogeneity associated with metabolic abnormalities, 732 subjects who did not have any component of MetS were set as the reference group and did not participate in the clustering, and 3900 samples with metabolic abnormalities were clustered into subtypes according to metabolic traits and covariates using multi-trait limited mixed regression (MFMR) [6].

The core assumption of MFMR is that subtypes differ in terms of distributions of several traits, resulting in a structure that can be identified by computational algorithms. This approach is more suitable for our study, since it accommodates differences in a variety of phenological traits and intrinsic genetic traits and can be easily adapted to both binary and quantifiable traits. In brief, MFMR assumes a single quantitative trait *y*, covariates *X*, discrete subtypes *z*, and a focal covariate *g* putatively interacting with *z*, and the model is:$$y_{i}=X_{i}\alpha +\gamma_{z_{i}}+g_{i}\beta_{z_{i}}+\mu_{i}$$

*X_i_* is a vector of Q control covariates, such as genetic PCs or sex, with homogeneous effect sizes *α*. *z_i_* is a K-level factor specifying the subtype for individual *i*, and *γ_k_* are the subtype main effects. *β* is the vector of subtype-specific *g* effect sizes. *g* is homogeneous if *β*_1_ = … = *β_K_*; otherwise, *g* is heterogeneous. Finally, *μ_i_* is i.i.d. Gaussian with mean zero. 

The key traits driving the subtypes in this study were continuous variables, including BMI, waist circumference, SBP, DBP, FBG, 2h-PBG, TC, TGs, HDL-C, and LDL-C. Covariates such as age, sex, smoking, drinking, the first three principal components, and medication history of antihypertensive drugs, antidiabetic drugs, and lipid-lowering drugs were adjusted for MFMR clustering.

#### 2.5.2. Characterizing Subtypes and Exploring the Associations between Subtypes and Cardiovascular Diseases

The characteristics of different subtypes were described after MFMR clustering. Categorical variables were presented as frequencies and percentages. Continuous variables with normal distributions were described using means and standard deviations, whereas those with skewed distributions were described using medians and upper and lower quartiles. Comparisons between multiple subtypes were performed using the χ^2^ test, analysis of variance (ANOVA) testing, or the Kruskal–Wallis test. Since taking drugs may obscure the true levels of biochemical indicators, resulting in the confusion of clustering results, the descriptions of the metabolic characteristics of subtypes were further stratified by medication.

The association of each subtype with CVD outcomes was estimated as an OR and a 95% CI by a logistic regression model with reference to the reference group. Variables in the multivariable models included age, sex, and number of MetS components diagnosed. Statistical significance was set at a *p*-value of 0.05. Analyses were conducted in R (v.4.1.2, R Core Team 2022, R Foundation for Statistical Computing, Vienna, Austria).

#### 2.5.3. Exploring the Potential Genetic Bases of the Subtypes

We conducted exome-wide association studies (EWASs) to identify risk loci for each metabolic subtype. The cases were subjects clustered into specific subtypes, and the controls were the reference group. Assuming an additive genetic model, we performed single-variant tests with a case–control study design, using logistic regression as implemented in PLINK 2.0 (a toolset for whole-genome association and population-based linkage analyses) [21]. Age, sex, and the first three principal components of the study population were included as covariates in the model to calculate odds ratios (ORs) and 95% confidence intervals (95% CIs). The genome-wide significance level for EWASs was defined as *p* < 7.5 × 10^−8^, and *p* < 5.0 × 10^−5^ was set to identify suggestive SNPs associated with subtypes [22,23].

To further gain biological insights into metabolic subtypes, suggestive SNPs identified by EWASs were mapped to genes according to their physical distances (an SNP was mapped to every gene whose coding sequence had an overlap with a 50 kb range around the SNP) and then tested for pathway and GTEx (Genotype–Tissue Expression) tissue-enrichment analysis using the MAGMA (Multi-marker Analysis of GenoMic Annotation) plug-in [24] on the FUMA (functional mapping and annotation) platform [25]. Enrichment of gene ontology functions and pathways was carried out by hypergeometric tests to recognize any statistical genetic over-representation from the input list (mapped from SNPs) in predefined MSigDB (Molecular Signatures Database) gene ontology gene sets (describing biological processes, molecular functions, and cellular components). MAGMA tissue expression analysis was performed using gene expression data from 83 tissues (30 general tissue types and 53 specific tissue types) based on GTEx RNA-seq data v7 [26]. Bonferroni-based multiple testing correction was applied for both analyses separately, by categories or tissue types. 

## 3. Results

### 3.1. Characteristics of Five Inferred Metabolic Subtypes

A total of 4655 subjects had phenotypic and genotypic data. After excluding subjects younger than 40 years old or missing key variables, 4632 individuals were finally included in the analysis. The average age was 57.1 ± 8.9 years, and 37.0% were male. A total of 1142 (24.7%) subjects were diagnosed with MetS. In terms of the prevalence of specific MetS components, 1477 (31.9%) had abdominal obesity, 2679 (57.8%) had hypertension, 1546 (33.4%) had hyperglycemia, 1416 (30.6%) had high blood TGs, and 479 (10.3%) had low blood HDL-C. Half of the study subjects had hypertension (50.4%) or hyperlipidemia (50.0%), a quarter (23.7%) had type 2 diabetes, and a few had CHD (10.9%) or stroke history (4.0%) (Table 1).

Among all the subjects, 732 (15.8%) individuals without any clinical diagnostic criteria for MetS were selected as the reference group and did not participate in MFMR clustering. In the clustering of the remaining 3900 individuals, the number of subtypes (K) was chosen according to the recommendation to maximize cross-validated likelihood, and the outputs of four clusters were determined (Appendix A). According to the differential distribution of metabolic characteristics, the subtypes were named: I, the adiposity-dominant type; II, the hypertension-dominant type; III, the hyperglycemia-dominant type; and IV, the dyslipidemia-dominant type, with sample sizes (proportions) of 428 (9.2%), 1617 (34.9%), 919 (19.8%), and 936 (20.2%), respectively. Specifically, subtype I was characterized by abdominal obesity, with a BMI of 28.5 ± 3.2 kg/m^2^ and a waist circumference of 90.2 ± 6.7 cm; subtype II was characterized by elevated blood pressure, with SBP of 143.7 ± 15.7 mmHg and DBP of 79.7 ± 10.3 mmHg; subtype III was characterized by high blood glucose, with FBG of 7.4 ± 2.5 mmol/L and 2h-PBG of 11.6 ± 4.9 mmol/L; and subtype IV was characterized by dyslipidemia, with TGs of 2.6 ± 1.8 mmol/L and HDL-C of 1.2 ± 0.3 mmol/L (Table 1, Figure 1). The prevalence of MetS components was also distinguished by subtypes, with abdominal obesity rates of 81.8% in subtype I, high blood pressure rates of 93.7% in subtype II, high blood glucose rates of 80.7% in subtype III, and high blood TGs and low HDL-C rates of 75.1% and 24.3% in subtype IV (Table 1, Figure 2). When further stratified by medication, the dominant metabolic characteristics of the subtypes of patients not taking antihypertensive drugs, antidiabetic drugs, or lipid-lowering drugs were more specific and more prominent than those taking these drugs (Appendix A).

### 3.2. Associations between Metabolic Subtypes and Cardiovascular Diseases

After accounting for potential confounding variables, such as age, sex, and number of MetS components diagnosed, the prevalence rates of CHD in all subtypes were significantly higher than in the reference group, with an OR (95% CI) of 6.440 (3.177–13.977) for III, the hyperglycemia-dominant subtype; an OR (95% CI) of 4.400 (2.314–9.119) for II, the hypertension-dominant subtype; an OR (95% CI) of 3.700 (1.729–8.370) for IV, the dyslipidemia-dominant subtype; and an OR (95% CI) of 3.461 (1.365–8.880) for I, the adiposity-dominant subtype. Furthermore, the dyslipidemia-dominant subtype and the hyperglycemia subtype were significantly associated with stroke (OR: 2.450, 95% CI: 1.250–5.265; 2.119, 1.063–4.614). However, the association between the other two subtypes and stroke was not statistically significant (*p* > 0.05) (Figure 3).

### 3.3. GO Pathway Enrichment and GTEx Tissue-Specific Enrichment for Metabolic Subtypes

After quality control and genotype imputation, the dataset contained 4632 samples and 669,504 SNPs. EWASs were performed separately for each subtype: obesity, hypertension, hyperglycemia, and dyslipidemia. We identified eight SNPs related to the dyslipidemia-dominant subtype with genome-wide significance. These sites were located in the genes *APOA5*, *BUD13*, *ZNF259*, and *WNT4* (Appendix A). In addition, some suggestive risk loci for the four subtypes were further used for pathway analysis and tissue enrichment (Appendix A).

In the abdominal obesity subtype, risk loci showed the enrichment of genes involved in the regulation of ion transmembrane transport pathways, and gene expression was significantly increased in heart regions, such as the left ventricle and the atrial appendage, and in brain regions, such as the cortex and amygdala. In the hypertension subtype, risk loci were enriched for genes involved in chaperone-mediated autophagy and dynactin complex pathways, and significant enrichment was found for genes expressed in heart regions, including the left ventricle and the atrial appendage, and in skeletal muscle. In the hyperglycemia subtype, regulation of JUN kinase activity and peptidyl serine phosphorylation were observed by pathway enrichment of mapped genes, and enrichment in genes expressed in the cortex and medulla of the kidney region, in the cerebellum and cerebellar hemisphere of the brain region, and in the atrial appendage and the left ventricle of the heart region were also highlighted, reaching significance. In the dyslipidemia subtype, gene-set analysis suggested the enrichment of risk genes in rap protein signal transduction and chaperone-mediated autophagy, and significant enrichment of expression in the left ventricle and atrial appendage of the heart region, the coronary and tibial arteries, breast mammary tissue, and skeletal muscle was observed (Figure 4 and Figure 5, Appendix A).

## 4. Discussion

Accurately characterizing the nature of each individual’s disease has significant implications for predicting outcomes, optimizing medical interventions, and discovering and correcting etiological mechanisms. To achieve these goals, disease subtypes are constantly being proposed, evaluated, and adopted in medical and scientific practice. Recent examples of proposed subtypes span diverse disease domains, including type 2 diabetes [3,27], breast cancer [28,29], asthma [30,31], autism [32,33], bipolar disorder, and schizophrenia [34,35]. However, few studies have paid attention to metabolic abnormalities as pre-disease states. Metabolic risk factors associated with serious and extensive comorbidities have been around for decades but tend to be clinically under-recognized [2]. Measuring the causative impacts of multi-dimensional and inter-related risk factors from a clustering perspective remains a major challenge, limiting opportunities for clinical translation. China is undergoing remarkable changes in lifestyle and socio-economic development, and it has one of the largest populations with metabolic abnormalities in the world. Based on an observational study of community residents over 40 years old in Beijing, this study identified metabolic subtypes from the perspective of clusters rather than single, affected individuals with respect to risk factors and further explored the association with CVD and the genetic basis of each subtype to gain an incisive view of metabolic heterogeneity.

We used MFMR clustering to identify four subtypes driven by metabolic characteristics. Subtype I was characterized by abdominal obesity, subtype II was characterized by hypertension, subtype III was characterized by hyperglycemia and had the highest CHD risk, and subtype IV was characterized by dyslipidemia and had the highest stroke risk. Unlike existing methods that computationally identify subtypes from high-dimensional trait data, MFMR accounts for covariates, especially population structures—important features of real genetic datasets [6]. The traits inputted into the clustering algorithm in this study were continuous variables, reflecting the metabolic statuses of individuals more comprehensively and subtly than categorical variables. The adjusted variables in the clustering algorithm in this study included age, sex, smoking, drinking, medication, and the first three principal components, which may confuse the subclassification but are not of interest to researchers. The results of such clustering are more likely to provide metabolic subtypes of genetic and pragmatic significance.

Recent studies have focused on the clustering of metabolic characteristics in an effort toward precision medicine for metabolism-related diseases. Chen et al. [36] found three distinct groups that presented significant differences in glucose and lipid metabolism. Meanwhile, molecular pathways that characterized the groups were also systematically described. Some researchers have been concerned with clustering metabolic features of hypertension patients. Vaura et al. [37] identified a metabolically challenged subgroup in the hypertensive population characterized by elevated blood glucose and BMI, with a higher risk of CVD. Yang et al. [38] observed four clusters in hypertensive patients: those in cluster 1 were relatively healthy, cluster 2 had a slight decrease in estimated glomerular filtration rate (eGFR), cluster 3 had the highest BMI, and cluster 4 had the highest Framingham risk score (FRS) for 10-year CVD risk. Cluster 4 also had the highest incidence of CVD outcomes, while the other groups did not differ from each other. Guo et al. [39] also observed four clusters in patients with hypertension: cluster 1 consisted mainly of younger male smokers, cluster 2 consisted mainly of older diabetic females, cluster 3 consisted mainly of relatively healthy individuals, and cluster 4 consisted mainly of individuals with coronary artery disease (CAD). There has also been some focus on the clustering of metabolic features in patients with diabetes. Ahlqvist et al. [3] divided diabetic patients into five groups based on six metabolic variables: cluster 1 was labeled as severe autoimmune type, cluster 2 as severe insulin-deficient type, cluster 3 as severe insulin-resistant type, cluster 4 as mild obesity-related type, and cluster 5 as mild age-related type. In particular, individuals in cluster 3 had a significantly higher risk of diabetic nephropathy than those in clusters 4 and 5 yet received similar treatment. Cluster 2 had the highest risk of retinopathy. Slieker et al. [40] conducted an external population validation and found similar results. Li et al. [41] observed a higher risk of other metabolic abnormalities in type 2 diabetic families, particularly hypertension and abdominal obesity. Notably, previous studies started with a certain metabolically relevant disease and investigated the role of metabolic risk factors in improving the etiology or prognosis through disease subcategories. This study attempts a new direction that considers the risk factors themselves in the general population and further explores associations with disease endpoints.

To further assess the biological relevance of each inferred metabolic subtype, we found several genetic variants that were strongly associated with the dyslipidemia-dominant subtype through EWASs. These sites were mapped to genes implicated in lipid metabolism, as suggested by previous genetic studies. To be specific, *APOA5* variants affect not only total TG concentrations but also the entire lipoprotein subclass distribution, shifting them toward atherogenic dyslipidemia in high-risk subjects [42]. The *BUD13*/*ZNF259* SNPs have been associated with one or more serum lipid traits in European populations, which associations are reproducible in Southern Chinese populations. Moreover, inter-locus interactions may exist among these SNPs [43]. An association of the *BUD13*-*ZNF259*-*APOA5*-*APOA1*-*SIK3* cluster polymorphism in 11q23.3 with increased plasma triglyceride levels has been reported in a Korean population [44]. However, for the other three subtypes, even the most significant SNPs could not reach the significance threshold for Bonferroni correction or FDR correction. Previous studies [45] have suggested that many small effect sites rather than a few large effect sites play an important role in metabolic traits, which places a requirement on sample sizes. The true associations of these sites with metabolic subtypes require larger sample sizes for validation.

Post-EWAS functional analyses provided insights into the underlying biological mechanisms of metabolic subtypes. In this study, subtypes with different metabolic characteristics and disease risks were identified at the phenotypic level, and further attempts were made to explore the genetic bases of subtypes at the locus, gene, and functional levels. Consistent with previous studies [46], the enrichment of genes involved in the regulation of ion transmembrane transport pathways was observed in obesity-dominant subtypes. The risk loci of hypertension-dominant subtypes were enriched for genes involved in chaperone-mediated autophagy and dynactin complex pathways, supporting previous reports that autophagy regulates angiotensin II-induced vascular smooth muscle cell hypertrophy [47]. The enrichment pathways of the hyperglycemia-dominant subtype indicated the role of c-Jun NH(2)-terminal kinase [48] and transferable phosphorus-containing groups [46] in the development of diabetes. The role of the signal transducer of rapeseed protein [49] and chaperone-mediated autophagy [50,51] has also been highlighted in this study and in previous studies. Among the top genes associated with metabolic subtypes, we observed enrichment for genes expressed in the heart and brain, supporting the involvement of cardiovascular disorders. Furthermore, previous studies have shown that variants for a particular disease appear to be enriched in disease-relevant cell types [52,53,54]. We found significant enrichment in the kidneys and arteries for the hyperglycemia-dominant subtype and the dyslipidemia-dominant subtype, respectively, suggesting possible etiological clues about organ damage.

There are some limitations to our study. First, all of the subjects included were Chinese. The applicability and stability of our classification with respect to external populations, particularly other ethnicities, requires further evaluation. Second, based on historical data from an observational study, residual measured and unmeasured confounders cannot be completely discounted. The predictive value of subtypes for the risk of cardiovascular disease should be verified as cohort data become available.

## 5. Conclusions

In conclusion, four metabolic subtypes were identified with the dominant characteristics of abdominal obesity, hypertension, hyperglycemia, and dyslipidemia. Furthermore, these subtypes appear to have different CVD risks and genetic bases. Our findings provide directions for future attempts at risk stratification and evidence-based management in populations with metabolic abnormalities from a systematic perspective.

## Figures and Tables

**Figure 1 biomedicines-10-03093-f001:**
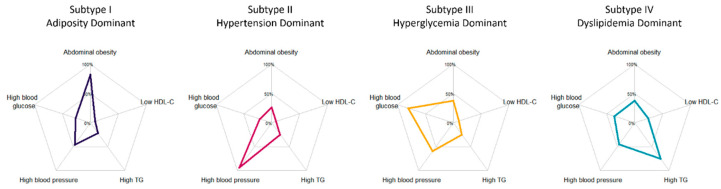
Prevalence of metabolic syndrome components in different subtypes *. * TG, triglyceride; HDL-C, high-density lipoprotein cholesterol.

**Figure 2 biomedicines-10-03093-f002:**
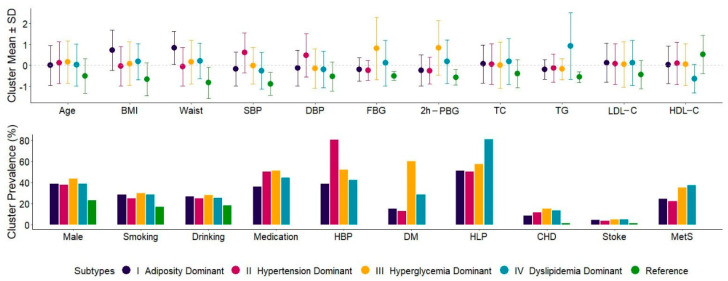
Quantitative and binary distributions for characteristics of different subtypes *. * Continuous variables are shown as means and standard deviations after being standardized to a unified scale. BMI, body mass index; Waist, waist circumference; SBP, systolic blood pressure; DBP, diastolic blood pressure; FBG, fasting blood glucose; 2h-PBG, 2 h postprandial blood glucose; TC, total cholesterol; TG, triglyceride; LDL-C, low-density lipoprotein cholesterol; HDL-C, high-density lipoprotein cholesterol; HBP, hypertension; DM, type 2 diabetes; HLP, hyperlipidemia; CHD, coronary heart disease; MetS, metabolic syndrome.

**Figure 3 biomedicines-10-03093-f003:**
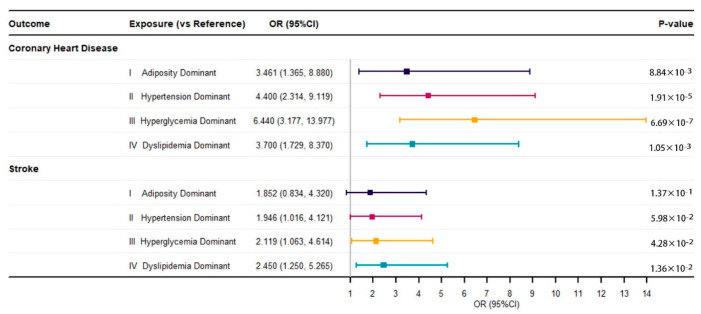
Estimates for the effects of metabolic subtypes on cardiovascular diseases *. * Odds ratios (ORs) and 95% confidence intervals (95% CIs) were estimated by multiple logistic regression models adjusted for age, sex, and number of MetS components diagnosed.

**Figure 4 biomedicines-10-03093-f004:**
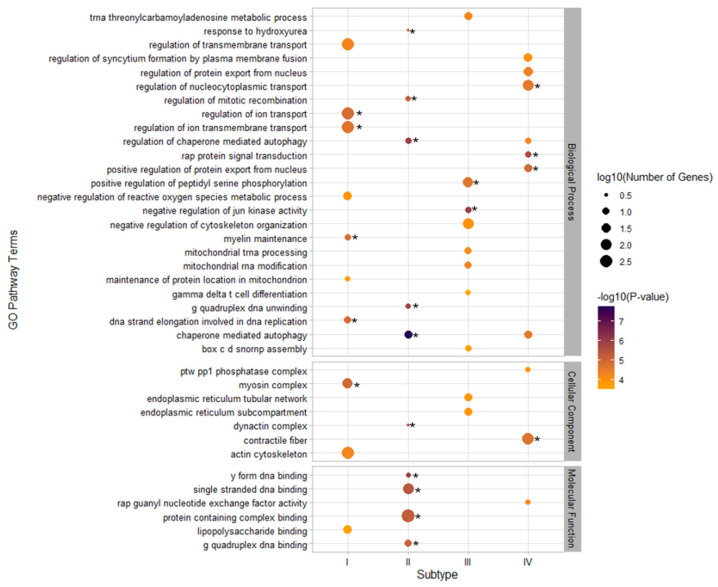
Gene ontology pathway enrichment analysis for metabolic subtypes ^◊^. ^◊^ Subtype I, adiposity-dominant type; Subtype II, hypertension-dominant type; Subtype III, hyperglycemia-dominant type; Subtype IV, dyslipidemia-dominant type. The top 10 gene sets for each subtype are shown, with significant gene sets marked with an “*” after conducting Bonferroni correction.

**Figure 5 biomedicines-10-03093-f005:**
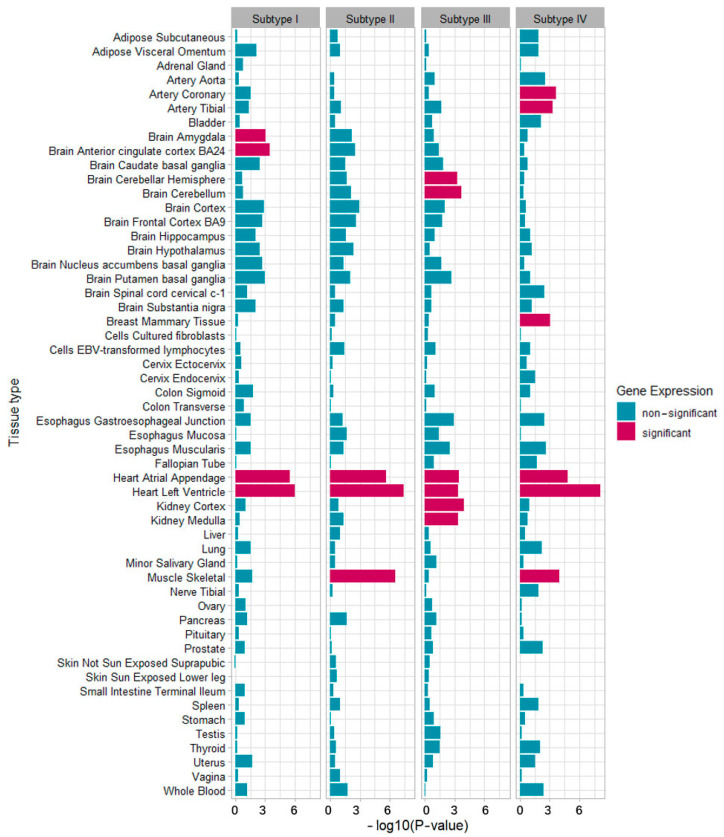
GTEx tissue-specific enrichment analysis for metabolic subtypes *. * Subtype I, adiposity-dominant type; Subtype II, hypertension-dominant type; Subtype III, hyperglycemia-dominant type; Subtype IV, dyslipidemia-dominant type.

**Table 1 biomedicines-10-03093-t001:** Characteristics of the studies with respect to five inferred metabolic subtypes *.

	Overall (*N* = 4632)	Subtype I (*N* = 428)	Subtype II (*N* = 1617)	Subtype III (*N* = 919)	Subtype IV (*N* = 936)	Reference (*N* = 732)	*p*-Values among Subtypes
Age, years, mean ± sd	57.1 ± 8.9	57.1 ± 8.6	58.2 ± 8.9	58.6 ± 9.0	57.4 ± 9.0	52.6 ± 7.5	0.003
BMI, kg/m^2^, mean ± sd	26.0 ± 3.4	28.5 ± 3.2	25.9 ± 3.2	26.3 ± 3.5	26.6 ± 2.9	23.8 ± 2.6	<0.001
Waist circumference, cm, mean ± sd	83.0 ± 8.5	90.2 ± 6.7	82.5 ± 7.8	84.4 ± 8.9	84.8 ± 7.2	76.0 ± 6.4	<0.001
SBP, mmHg, mean ± sd	133.7 ± 16.5	130.9 ± 13.3	143.7 ± 15.7	133.5 ± 14.6	129.5 ± 14.4	119.1 ± 9.2	<0.001
DBP, mmHg, mean ± sd	74.9 ± 9.9	73.6 ± 8.5	79.7 ± 10.3	73.3 ± 9.3	72.9 ± 8.7	69.6 ± 7.0	<0.001
FBG, mmol/L, mean ± sd	6.1 ± 1.7	5.8 ± 0.9	5.7 ± 0.8	7.4 ± 2.5	6.3 ± 1.8	5.3 ± 0.4	<0.001
2h-PBG, mmol/L, mean ± sd	8.4 ± 3.8	7.5 ± 2.8	7.5 ± 2.4	11.6 ± 4.9	9.1 ± 4.0	6.3 ± 1.4	<0.001
TC, mmol/L, mean ± sd	5.3 ± 1.0	5.4 ± 0.9	5.4 ± 1.0	5.3 ± 1.1	5.5 ± 1.1	4.9 ± 0.7	0.002
TG, mmol/L, mean ± sd	1.6 ± 1.1	1.4 ± 0.5	1.4 ± 0.8	1.4 ± 0.6	2.6 ± 1.8	0.9 ± 0.3	<0.001
LDL-C, mmol/L, mean ± sd	3.2 ± 0.8	3.3 ± 0.8	3.3 ± 0.8	3.3 ± 0.9	3.3 ± 0.9	2.9 ± 0.6	0.342
HDL-C, mmol/L, mean ± sd	1.4 ± 0.4	1.4 ± 0.3	1.5 ± 0.4	1.4 ± 0.4	1.2 ± 0.3	1.6 ± 0.3	<0.001
Male, *n* (%)	1713 (37.0%)	166 (38.8%)	615 (38.0%)	399 (43.4%)	362 (38.7%)	171 (23.4%)	0.054
Smoking, *n* (%)	1192 (25.7%)	122 (28.5%)	401 (24.8%)	276 (30.0%)	269 (28.7%)	124 (16.9%)	0.019
Drinking, *n* (%)	1153 (24.9%)	114 (26.6%)	407 (25.2%)	259 (28.2%)	237 (25.3%)	136 (18.6%)	0.370
Medication, *n* (%)							
Antihypertensive drugs	1488 (32.1%)	115 (26.9%)	736 (45.5%)	303 (33.0%)	334 (35.7%)	-	<0.001
Antidiabetic drugs	537 (11.6%)	52 (12.1%)	127 (7.9%)	211 (23.0%)	147 (15.7%)	-	<0.001
Lipid-lowing drugs	495 (10.7%)	64 (15.0%)	173 (10.7%)	130 (14.1%)	128 (13.7%)	-	<0.001
Hypertension, *n* (%)	2335 (50.4%)	166 (38.8%)	1296 (80.1%)	479 (52.1%)	394 (42.1%)	-	<0.001
Type 2 diabetes, *n* (%)	1096 (23.7%)	66 (15.4%)	212 (13.1%)	551 (60.0%)	267 (28.5%)	-	<0.001
Dyslipidemia, *n* (%)	2317 (50.0%)	218 (50.9%)	816 (50.5%)	525 (57.1%)	758 (81.0%)	-	<0.001
Coronary Heart Disease, *n* (%)	505 (10.9%)	37 (8.6%)	192 (11.9%)	140 (15.2%)	125 (13.4%)	11 (1.5%)	0.004
Stroke, *n* (%)	185 (4.0%)	19 (4.4%)	63 (3.9%)	45 (4.9%)	48 (5.1%)	10 (1.4%)	0.444
Metabolic syndrome, *n* (%)	1142 (24.7%)	105 (24.5%)	365 (22.6%)	322 (35.0%)	350 (37.4%)	-	<0.001
Metabolic syndrome components, *n* (%)							
Abdominal obesity	1477 (31.9%)	350 (81.8%)	424 (26.2%)	349 (38.0%)	354 (37.8%)	-	<0.001
High blood pressure	2679 (57.8%)	195 (45.6%)	1515 (93.7%)	548 (59.6%)	421 (45.0%)	-	<0.001
High blood glucose	1546 (33.4%)	112 (26.2%)	349 (21.6%)	742 (80.7%)	343 (36.6%)	-	<0.001
High blood TG	1416 (30.6%)	93 (21.7%)	395 (24.4%)	225 (24.5%)	703 (75.1%)	-	<0.001
Low blood HDL-C	479 (10.3%)	37 (8.6%)	121 (7.5%)	94 (10.2%)	227 (24.3%)	-	<0.001

* BMI, body mass index; SBP, systolic blood pressure; DBP, diastolic blood pressure; FBG, fasting blood glucose; 2h-PBG, 2 h postprandial blood glucose; TC, total cholesterol; TG, triglyceride; LDL-C, low-density lipoprotein cholesterol; HDL-C, high-density lipoprotein cholesterol; TIA, transient ischemic attacks.

## Data Availability

Not applicable.

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
