# Peer review of "Identification of Novel Metabolic Subtypes Using Multi-Trait Limited Mixed Regression in the Chinese Population"

_biomedicines, 2022, doi:10.3390/biomedicines10123093_

Round 1
Reviewer 1 Report
In the present work, the authors aimed at three main goals. They have used community observational study data from Beijing, China, along with genotyping in order to a) infer metabolic subtypes using multi-trait limited mixed regression (MFMR) clustering approach from metabolic quantitative traits among the general population, b) investigate the association of inferred subtypes with coronary heart disease (CHD) and stroke, and c) explore the underlying genetic basis of inferred subtypes.
Their work is interesting and it has merit for publication, with some issues however. At first sight, it seems that this approach attempts to identify new types of associations between CHD, stroke and underlying subtypes. The authors reported that that risk subtypes and groups are adiposity, blood pressure, hyperglycemia, dyslipedemia, which are already known that are major risk factors for CHD and stroke (as well as for diabetes). Why is such an elaborate analysis required for something that can discovered using "classic" statistics such as OR, ANOVA, etc. Please comment on that.
The authors report on specific SNPs (I guess found in the population under investigation). Are these exclusive for the population under investigation? are they present in other populations, as for example Europe or the Americas? Which are these SNPs?
What was difficult to understand is how their approach infers risk factors? this could be a successful approach, if the population was divided into samples that did not manifest a certain disease at the start of the study (2005 if I got it correct) and then due to phenotypical factors developed a CHD or metabolic disease in the future (or at least up to the point of the present study). Thus, why was not the population divided to pre-disease and post-disease subjects? That way the authors could possibly find risk factors other than the "usual suspects". Please comment on that.
Final, the authors should summarize in their conclusions how their approach (and this overall approach) can help identify risk factors, that appear early, before the actual disease hits.
Author Response
Dear reviewer,
We are pleased to send a revised manuscript and a point-by-point response for your re-evaluation. Please see the attachment.
The comments are all valuable and helpful for improving our article. All the authors have seriously discussed these comments and have tried our best to revise the manuscript. In this revised version, changes were highlighted.
If there are still some questions in the revised version, we will be very pleased to address them. We highly appreciate your further consideration of this manuscript.
Yours faithfully,
Dafang Chen, MD
Peking University
No 38, Xueyuan Rd., Haidian District, Beijing, China, 100191
Tel: +86-10-82802644
E-mail: dafangchen@bjmu.edu.cn

Reviewer 2 Report
Kexin Ding et al attempt to understand risk stratification considering metabolic risk factors. They use available dataset and establish major subtypes and correlate these with coronary heart disease, stroke; they identify risk SNPs and highlight major pathway enrichments and gene locations from exome sequencing data. Overall the study is fairly well done and very compelling in the field; however some clarifications are needed and additional data must be added.
· It would be very important that authors identify SNPs related to the other subtypes, like the ones they found for dyslipidemia.
· Authors should clarify better the rationale to look at pathway enrichments and at the location where the significant genes are predicted to be expressed. Authors should include literature where tissue type expression and its importance is highlighted when doing predictions and these kind of correlation studies. What are authors trying to infer when stating locations for enriched genes in the different subtypes?
Author Response

(The authors gave the same response as above.)

Reviewer 3 Report
The present study about the “Identification of novel metabolic subtypes using multi-trait limited mixed regression in the Chinese population” is interesting and provides information ofonour metabolic subtypes which were identified, with the dominant characteristics of abdominal obesity, hypertension, hyperglycemia, and dyslipidemia. Further providing evidence for future attempts at risk stratification and evidence-based management in populations with metabolic abnormalities from a systematic perspective. However, the authors should address the following minor questions.
· Authors are suggested to expand the acronym MFMR.
· Authors are suggested to expand all the acronyms when they first appear in the text. For ex: SHAPEIT v2.
Author Response

(The authors gave the same response as above.)

Round 2
Reviewer 1 Report
The authors have addressed my previous comments. Their manuscript can be published in its present form.
Author Response
Dear reviewer,
On behalf of all the authors, thank you very much for agreeing with the publication of this article. Thank you again for your valuable review, which helped us greatly improve the quality of our article.
Yours faithfully,
Dafang Chen, MD
Peking University
No 38, Xueyuan Rd., Haidian District, Beijing, China, 100191
Tel: +86-10-82802644
E-mail: dafangchen@bjmu.edu.cn
Reviewer 2 Report
Authors have provided clarifications and improved the manuscript.
One point not answered in the first round:
· It would be very important that authors identify SNPs related to the other subtypes, like the ones they found for dyslipidemia.
Author Response
Dear reviewer,
Thank you for your valuable suggestion, which is essential to improve our manuscript.
We reported and discussed 8 SNPs related to the dyslipidemia dominant subtype with genome-wide significance (P < 5×10-8), which could not avoid possible false-positive associations, so the authors did not specify these sites in the manuscript section, but mentioned them in Supplementary Table 4.
In Supplementary Table 4, we listed 327 suggestive SNPs that may contribute to metabolic subtypes (P < 5×10-5). The metabolic subtype is a polygenic trait with a complex genetic structure. Previous studies have suggested that many small effect sites rather than a few large effect sites play an important role in metabolic traits, which places a requirement on sample sizes. The true association of these sites with metabolic subtypes requires larger sample sizes to validate. We have added the relevant description in the discussion section.
If there are still some questions in the revised version, we will be very pleased to address them. We highly appreciate your further consideration of this manuscript.
Yours faithfully,
Dafang Chen, MD
Peking University
No 38, Xueyuan Rd., Haidian District, Beijing, China, 100191
Tel: +86-10-82802644
E-mail: dafangchen@bjmu.edu.cn